# OPEN SET DOMAIN ADAPTATION WITH ZERO-SHOT LEARNING ON GRAPH

## ABSTRACT

Open set domain adaptation focuses on transferring the information from a richly labeled domain called *source domain* to a scarcely labeled domain called *target domain* while classifying the unseen target samples as one *unknown* class in an unsupervised way. Compared with the close set domain adaptation, where the source domain and the target domain share the same class space, the classification of the unknown class makes it easy to adapt to the realistic environment. Particularly, after the recognition of the unknown samples, the robot can either ask for manually labeling or further develop the classification ability of the unknown classes based on pre-stored knowledge. Inspired by this idea, in this paper we propose a model for open set domain adaptation with zero-shot learning on the unknown classes. We utilize adversarial learning to align the two domains while rejecting the unknown classes. Then the knowledge graph is introduced to generate the classifiers for the unknown classes with the employment of the graph convolution network (GCN). Thus the classification ability of the source domain is transferred to the target domain and the model can distinguish the unknown classes with prior knowledge. We evaluate our model on digits datasets and the result shows superior performance.

## 1 INTRODUCTION

In the last decades, deep learning models have shown good performance in various tasks, especially in visual perception. The training of the deep learning network relies on plenty of labeled data. However, most of the existing large labeled datasets are collected from the Internet. The images in these datasets are normative and unified, which are different from the images relevant for a specific application. Besides, depending on the application, the images may be obtained by different typed of visual sensors or with a different perspective of sensors. It costs a lot to retrain the classification model in different situations. In some typical applications, the samples in the real world are hard to gather or too large to label. Thus it is important to deal with the gap among domains. They should be able to utilize the well-labeled samples in the source domain to classify the samples in the unlabeled target domain which is related to domain adaptation. There are already some researches on domain adaptation, such as Ganin & Lempitsky (2015), Long et al. (2015), Long et al. (2016), and Wang & Deng (2018). The alignment of the domain gap makes the robot adapt well to dynamic and unstructured environments.

Except for the domain gap among different datasets, the variation of the classes also makes it hard for the model to adapt to a new dataset. Depending on the application and the scale of different datasets, the model may come across classes that are not contained in the source domains. With the traditional domain adaptation methods, the unknown classes are mistakenly aligned due to the absence of training samples of unknown classes in the source domain. The imbalance of the types of classes brings over-fitting problems and is not suitable for classification in the open world. Thus it is important for the robot to reject the unknown classes and only align the shared classes. This problem is known as open set domain adaptation, which is first proposed by Panareda Busto & Gall (2017) and followed by for instance Saito et al. (2018), Busto et al. (2018), and Liu et al. (2019). In the setting of the open set domain adaptation, the target domain contains both the classes of the source domain and the additional new classes. The model not only aligns the target domain to the source domain but also rejects the unknown classes.

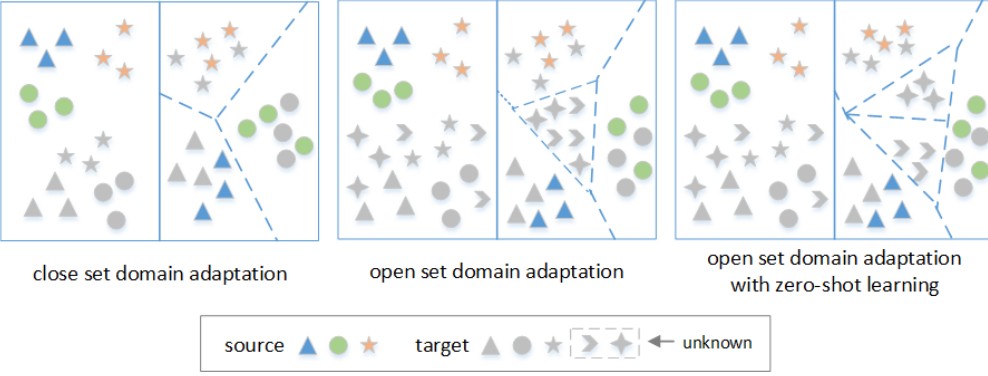

Figure 1: An overview of the proposed domain adaptation with zero-shot learning. Close set domain adaptation align the target domain to the source domain. Open set domain adaptation not only align the domain gap, but also reject the unknown classes as one class. Open set domain adaptation with zero-shot learning further gives detailed classification on the unknown classes, which is more complex and valuable.

It is worth noting that previous open set domain adaptation methods typically classify all the additional new classes into one *unknown* class. However, the unknown class may contain classes that are worth learning. It may be more valuable to detect the unknown classes in detail and develop the ability to classify them with the former information. With the process of distinction and transferring, the model can expand its visual recognition ability with little labeled information. Since the unknown classes are not included in the source domain, the model lacks the labeled information for the new classes. Current open set domain adaptation methods can not give detailed classification on the unknown part with no labeled images. This problem is related to zero-shot learning. In the zero-shot learning problem, complementary information is collected to transfer the knowledge from the base classes to classify the unknown ones. Inspired by this, with the knowledge stored in the knowledge graph, the classifiers of the unknown classes can be obtained in the target domain with no labeled samples.

Towards this end, we propose a generic model to align the gap between the labeled source domain and the unlabeled target domain while classifying the unknown classes in the target domain. The contributions of this paper mainly lies in tackling the following two difficulties.

First, since the unknown classes are not contained in the source domain, we have no labeled samples for supervised training. The lack of labeled data may cause overfitting problem of the model, which means the model only classify the samples as the known classes and can not classify the unknown ones. It is necessary to utilize complementary information to support the inference. Thus we employ the knowledge graph to stores some prior knowledge of the known classes and the unknown classes, which contains the structural relations between different classes, beyond the individual attribute representation of each class. The structural information offers a bridge for the inference from the known classes to the unknown ones. With the employment of the graph convolution network, the information propagates among the graph and the unknown classes gather the information from their neighbor to generate their classifiers. These inference classifiers work as the initial classifiers of the classification model.

The second difficulty is how to adapt the inference classifiers to the target domain. Since we only have labeled samples in the source domain, the inference classifiers are suitable to the source domain. It is not able to classify the unknown samples in the target domain because of the domain gap. Thus we introduce adversarial learning to align the domain gap. The classification model consists of two modules, the feature generator, and the classifier. Since the generator works to extracts the features of the samples and the classifier works to output the class probability, we train them simultaneously in an adversarial way. The classifier is trained to found a boundary for the unknown classes while the generator is trained to make the samples far from the boundary. With adversarial learning, the generator can deceive the classifier to generate aligned features in both domains and reject the unknown classes according to the unknown boundary. Thus the feature of the shared classes

is aligned in both domains and the unknown classes are rejected as one class. With the adaptation in both domain gap and class gap, our model is able to classify objects in the dynamic and complex open world.

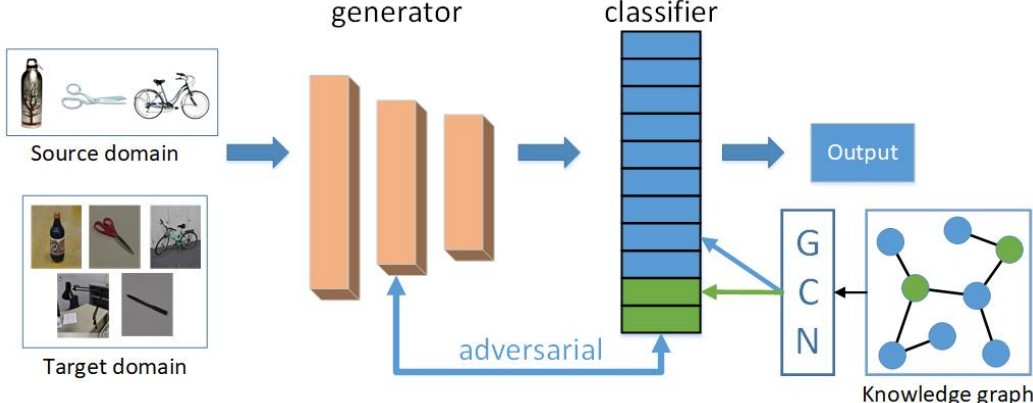

Figure 2: An overview of our model for domain adaptation with zero-shot learning. We utilize a knowledge graph to infer the classifier of the unknown classes. With the employment of the graph convolutional network, the model generates the initial classifier for all classes and applies it to the classification part. The generator and the classifier are trained in an adversarial way to align the domain gap.

We utilize the knowledge graph and the adversarial learning in a jointly trained framework. The two parts work together to align the shared classes in two spaces while generating classifiers for the unknown classes in the target domain. We further evaluate our method on digits datasets and demonstrate its effectiveness.

## 2 RELATED WORKS

### 2.1 OPEN SET DOMAIN ADAPTATION

Open set domain adaptation goes beyond traditional close set domain adaptation. It considers a more realistic classification task, in which the target domain contains unknown samples that are not present in the source domain. Open set domain adaptation is first proposed by Busto et al. (2018). They measure the distance between the target sample and the center of the source class to decide whether a target sample belongs to one of the source classes or the unknown class. However, they require the source domain to have unknown samples as well. Later on, Saito et al. (2018) propose open set back-propagation (OSBP) for source domain with no unknown samples. They utilize adversarial learning to train the feature generator and classifier. As the classifier tries to set a boundary for the unknown classes, the feature generator tries to deceive it. However, both of them only separate the unknown classes in the target domain, but can not give detailed classification on the unknown ones. The learnable information in the unknown space deserves deep exploitation. We have found few papers that consider the fine-grained classification of the unknown classes in open set domain adaptation, we aim to fill in the blanks.

### 2.2 ZERO-SHOT LEARNING

Zero-shot learning aims at generating classifiers for unknown classes with no labeled samples. Several pieces of research have been done on this area, such as Kipf & Welling (2016) Xian et al. (2017). Due to the limitation of the available samples, some researchers extract complementary information from the related known classes to support the inference of the unknown ones. Among these methods, building the relationship between classes in form of a graph seems more reasonable. The special geometry of graphs well shows the complicated relationship and the unknown classes can gather adequate information from the known ones. Current zero-shot learning please refer to

knowledge graphs for inference. Wang et al. (2018) built an unweighted knowledge graph combined with word embedding upon the graph convolutional network. With information propagation, novel nodes generate predictive classifiers with common sense. Kampffmeyer et al. (2019) improve upon this model and propose a dense graph propagation to prevent dilution of knowledge from distant nodes. Knowledge graphs intuitively present the stored information for the visual cognitive development.

## 2.3 GRAPH CONVOLUTIONAL NETWORK

Graph convolutional network (GCN) is a kind of graph neural network proposed by Kipf & Welling (2016). GCN enables the nodes in the graph to share the intensity of statistics, which improves the efficiency of sampling. GCN can also be employed in non-euclidean space. From the perspective of the spatial domain, GCN iteratively aggregates neighborhood information. The propagation of information on the graph further exploits the structural information on the graph. GCN is first introduced by Bruna et al. (2013). Then GCN is extended with a filtering method based on the recurrent Chebyshev polynomial. The improvement reduces the computational complexity a lot, which is equivalent to common CNNs in image operating. Kipf & Welling (2016) further simplify the framework to improve the scalability and robustness. They employed their model on the semi-supervised learning on the graphs. Our model employs the framework of GCN to propagate the complementary information among nodes for the inference of classifiers.

## 3 APPROACH

### 3.1 PROBLEM DEFINITION

In open set domain adaptation with zero-shot learning, we have a source domain $D_s = \{(x_i^s, y_i^s)\}_{i=1}^{n_s}$, which contains $n_s$ labeled samples, and a target domain $D_t = \{x_j^t\}_{j=1}^{n_t}$, which contains $n_t$ unlabeled samples. The class space in the source domain is $C_s$ which we call known classes. $C_s$ also is shared by the class space of the target domain $C_t$. It is worth noting that $C_t$ further contains additional unknown classes $C_u$, that is $C_t = C_s \vee C_u$. The source domain is sampled from the distribution $q_s$, while the target domain is sampled from the distribution $q_t$. In close set domain adaptation, $q_s \neq q_t$. In open set domain adaptation with zero-shot learning, we also define $q_s \neq q_t^s$. $q_t^s$ refers to the distribution of the known classes in the target domain. Note that the samples in the target domain are all unlabeled and the samples in the source domain are all labeled.

### 3.2 CLASSIFIER INFERENCE MODULE

With few labeled samples, the human can make good inferences on unfamiliar things with the related information that they obtain from books. Our model also extracts the task-based knowledge from a prestored knowledge graph. The inference graph is denoted as $G = (V, E)$, where $V = \{v_1, v_2, ..., v_{n_s}, ..., v_{n_t}\}$ is a node-set of all classes $C_t$. $n_s$ refers to the number of known classes. $n_t - n_s$ refers to the number of unknown classes. The nodes in the prestored knowledge contain the attributes $v_i$ of different classes. $E = \{e_{i,j} = (v_i, v_j)\}$ is an edge set referring to the relationship among the graph. The edges in the prestored knowledge graph are based on the similarity of the attributes between different classes.

Since we only have labeled data of the source domain. We first train the recognition model on the source domain $D_s$. Specifically, the pre-trained recognition model is denoted as $C(F(\cdot|\theta)|w)$. The model consists of two parts, feature extractor $F(\cdot|\theta)$ and class classifier $C(\cdot|w)$. $\theta$ and $w$ indicate the parameters of the model trained with $D_s = \{(x_1, y_1), ..., (x_{n_s}, y_s)\}$. The symbol $x_i$ refers to the source images of the ith class while $y_i$ refers to their label. Feature extractor $F(x|\theta)$ takes an image as input and figures out the feature vector of it as $z_i$. The final classification score is computed as

$$[s_1, s_2, ..., s_M] = [z^T w_1, z^T w_2, ..., z^T w_{n_s}] \tag{1}$$

Thus the inference of the classifiers on unknown classes is to inference the classification weights $w_s$ on the unknown classes with the inference graph.

With the framework of the graph convolutional network, Our model propagates information among nodes by exploring the class relationship. For one layer in GCN, a node aggregates information

from the neighbors connected to it. GCN can also be extended to multiple layers to perform a deeper spread. Therefore, the unknown classes can utilize the information from the related known classes and predict the classification weights of their own. The mechanism of GCN is described as

$$H^{(l+1)} = ReLu(\hat{D}^{-\frac{1}{2}}\hat{E}\hat{D}^{-\frac{1}{2}}H^{(l)}U^{(l)}) \tag{2}$$

where $H^{(l)}$ denotes the output of the $l^{th}$ layer, while for the first layer $H^0 = V$. It uses Leaky ReLu as the nonlinear activation function. To reserve the self-information of the nodes, self-loops are added among the propagation, $\hat{E} = E + I$, where $E \in R^{N \times N}$ is the symmetric adjacency matrix and $I \in R^{N \times N}$ represents identity matrix. $D_{ii} = \sum_j E_{ij}$ normalizes rows in E to prevent the scale of input modified by $E$. The matrix $U^l$ is the weight matrix of the $l^{th}$ layer, which GCN regulates constantly to achieve better performance.

Our model conducts two layers of GCN on the inference graph. Unknown classes learn the mechanism of end-to-end learning from known classes through propagation. The inference graph is trained to minimize the loss between the predicted classification weights and the ground-truth weights. The ground-truth weights refer to the classifiers of the known classes, which are extracted from the pretrained model on the source domain.

$$L_{GCN} = \frac{1}{M}\sum_{i=1}^{M}(w_i^{inf} - w_i^{train})^2 \tag{3}$$

where $w^{inf}$ refers to the output of known classes on GCN. $w^{train}$ denotes the ground truth classifiers of the known classes obtained from the pre-trained model. With the supervision of the known classes, the unknown nodes in the inference graph can also generate classifier weights of their own. Finally, with the employment of GCN, the classifier inference module not only generates predictive classifiers of the unknown classes in the source domain but also provides more general classifiers of the known ones.

### 3.3 DOMAIN ADAPTATION MODULE

The classifier inference module generates classifiers for the unknown classes. However, these classifiers are suitable to the source domain, since the ground-truth classifiers are extracted from the model trained on the labeled samples. Thus the domain adaptation module attempts to align the domain gap between the source domain and target domain.

We apply the inference classifiers to the pre-trained classification model. The number of the classifiers of the pre-trained classification model changes from $n_s$ to $n_t$. As mentioned above, the classification model consists of two parts, the feature generator, and the classifier. To align the domain gap, we employ adversarial learning on the classifier and the feature generator. The classifier aims at setting a boundary for the unknown classes in the target domain. With the boundary, the unknown classes can be picked out. The boundary refers to the proportion of the unknown classes in the target domain.

$$p_{un} = p(y = y_{un}|x_t) = t \tag{4}$$

$$y_{un} = \sum_{i=n_s+1}^{n_t} p(y = y_i|x_t) \tag{5}$$

where $y_{un}$ refers to the proportion of unknown classes, $t$ refers to the boundary. The feature generator tries to generate features that can deceive the classifier. That is, the objective of the generator is to maximize the error of the classifier. To increase the error, the generator tries to generate features far from the boundary. Besides, the classification ability on the known classes should be reserved, thus we also consider the classification accuracy on the source domain during the training process. We use a standard cross-entropy loss for this purpose.

$$L_s(x_s, y_s) = -\log(p(y = y_s|x_s)) \tag{6}$$

$$p(y = y_s|x_s) = (C(F(x_s)))_{y_s} \tag{7}$$

With the cross-entropy loss, the model ensure the classification accuracy on known classes. For the boundary of the unknown classes, we follow the settings in the OSBP and utilize binary cross

entropy loss.

$$L_{adv}(x_t) = -t \log(p_{un}) - (1 - t) \log(1 - p_{un})) \tag{8}$$

$$p_{un} = \sum_{i=n_s+1}^{n_t} p(y = y_i | x_t) \tag{9}$$

The overall objective of our model is,

$$\min_C L_s(x_s, y_s) + L_{adv}(x_t) + L_{GCN} \tag{10}$$

$$\min_G L_s(x_s, y_s) - L_{adv}(x_t) + L_{GCN} \tag{11}$$

With the domain adaptation module, the unknown classes in the target domain are separated and the features of both domains are aligned. We also suggest iterating the classifier inference module and the domain adaptation module for better performance.

## 4 EXPERIMENT

In this section, we perform experiments to evaluate the effectiveness of our model. Since there is no related model on the open set domain adaptation problem with zero-shot learning, we demonstrate our model with current zero-shot learning models and make a detailed analysis of the results.

### 4.1 DATASETS

We test our model on three digits datasets. Compared to the traditional dataset on domain adaptation. The digits datasets contain a fewer number of classes, which means the number of the known classes is fewer. Thus the information that the unknown classes can utilize is fewer. The task turns to a more difficult zero-shot learning problem. The three digits datasets are MNIST, USPS, and SVHN. MNIST is proposed by LeCun et al. (1998), which contains about seventy thousand images of the white digits on a black background. USPS is proposed by Friedman et al. (2001), which contains seven thousand grayscale images on the handwritten digits. SVHN is proposed by Netzer et al. (2011), which contains six hundred thousand images of real-world street view house numbers. We consider each dataset as a different domain. For the class space, we have two settings of unknown classes. In the 3-way setting, the source domain contains seven classes (0-6), while the target domain contains ten classes (0-9). While in the 2-way setting, the source domain contains seven classes (0-7), while the target domain contains ten classes (0-9). In both settings, our goal is to align the known classes in the target domain to the source domain and have the ability to classify the unknown ones.

### 4.2 SETTINGS

We employ GCN of two layers for the inference of the classifier. The ground truth of the classifiers on the known classes is extracted from the original recognition model trained on the source domain. The original model has almost the same structure as the final classification model, except for the number of the classifier. The classification model is trained in 1200 epochs. We use the Adam Da (2014) optimizer for training with the weight decay of 0.0005 and the learning rate of 0.001. The boundary of unknown classes is set to 0.5 for better performance. The whole project is under the framework of PyTorch Paszke et al. (2017).

### 4.3 COMPARISON

To test the performance of our model, we conduct experiments under several settings. However, since there are few models that work on the domain adaptation with zero-shot learning, we compare our model to zero-shot learning methods. Besides verifying the value of the inference classifiers, we also compare our model to open set domain adaptation methods with random initialization unknown classifiers. The results are shown in the following table.

The comparison between our model and other exiting methods is reported in table 1. The task names s2m, u2m, and m2u refer to the transfer tasks from SVHN to MNIST, USPS to MNIST, and MNIST to USPS. The 2-way and 3-way settings mean the number of unknown classes is 2 and 3. The class

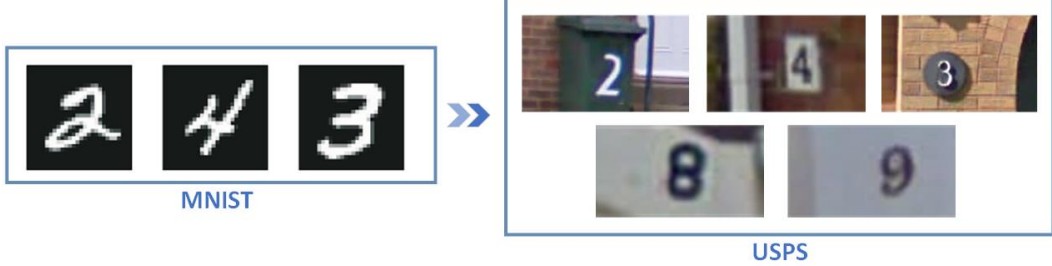

Figure 3: In the task setting m2u, the model is trained on the MNIST dataset with seven or eight classes and transferred to the USPS dataset with ten classes.

type setting all and unknown refers to the classification accuracy on the overall 10 classes and the unknown classes. The performance is evaluated by the average top-1 accuracy.

z-GCN proposed by Wang et al. (2018) is a zero-shot learning model with the employment of graph knowledge. It only considers the classes gap while ignores the domain gap. However, as figure 3 shows, the domain gap between the MNIST and the USPS is large. Overfitting on the source domain and the lack of labeled training images in the target domain affect the results a lot. In the s2m tasks with the 2-way setting, the classification accuracy of z-GCN on all classes is forty-eight, which is twenty percent lower than our model. In the u2m and m2u tasks, the improvement of our model is about ten percent as well. The result demonstrates that the adversarial learning employed by our model is able to transfer the classification ability from the source domain to the target domain with no labeled data. The domain adaptation is important for the flexibility of the models

Table 1: Comparison results

| task | s2m | | | |
|---|---|---|---|---|
| setting | 2-way | | 3-way | |
| classes | all | unknown | all | unknown |
| z-GCN | 48.4% | 6.2% | 39.5% | 13.2% |
| OSBP | 58.2% | 17.4% | 54.0 % | 24.3% |
| our | 67.0% | 38.2% | 64.3% | 46.4% |
| task | u2m | | | |
| setting | 2-way | | 3-way | |
| classes | all | unknown | all | unknown |
| z-GCN | 60.5% | 9.5% | 54.1% | 10.4% |
| OSBP | 61.4% | 8.5% | 62.3 % | 12.4% |
| our | 67.4% | 26.5% | 69.2% | 24.1% |
| task | m2u | | | |
| setting | 2-way | | 3-way | |
| classes | all | unknown | all | unknown |
| z-GCN | 63.4% | 8.6% | 62.0% | 12.3% |
| OSBP | 51.3% | 7.6% | 42.2% | 10.3% |
| our | 73.6% | 49.3% | 68.2% | 23.5% |

Although z-GCN aims at the zero-shot problem, it is worth noticing that the accuracy of the unknown classes in the target domain adaptation is around ten percent. The classification accuracy is low on the zero-shot learning problem. That is because the z-GCN only considers the gap in the class space, but does not take the domain gap into consideration. With the training process, z-GCN only infers the classifier for the source domain. However, the testing is performed on the target domain. This result shows the importance of domain adaptation in the zero-shot learning problem. With the aligning of the domain gap, the accuracy of the unknown classes is increased by about twenty percent in our model. In the s2m tasks, the improvement is about thirty percent on the unknown classes. The high classification accuracy on the unknown classes also confirms that the domain

adaptation not only aligns the known classes in both domains but also extracted the features of the images, which is domain invariant.

To test the effectiveness of the inference classifiers, we further conduct experiments on the open set domain adaptation method OSBP proposed by Saito et al. (2018). OSBP transfers the classification ability from the source domain to the target domain and rejects all of the unknown classes as one class. Since we aim at classifying the unknown classes in detail, we expand the OSBP with randomly initialized classifiers on the unknown classes. From the results in table 1, the classification ability on the unknown classes is about twenty percent lower than our method. However, we notice that the OSBP still shows about ten percent accuracy on the unknown classes and sometimes even higher than z-GCN. We owe the classification accuracy on the unknown classes to the rejection mechanism. Since OSBP has the ability to reject the unknown classes as one class, the detailed classification in the one class is much easier. The randomly initialized classifier still has the possibility to classify the two or three unknown classes properly.

The accuracy on unknown classes for our method is about twenty percent higher than OSBP. Besides, the accuracy on all classes still has about a ten percent increase. The result shows that the inference from the word embedding to the unknown classifier makes sense. Besides, the inaccurate classifiers on the unknown classes confuse the classifier on the known classes and result in a decrease in the accuracy. Thus the outperformance of our model is not simply due to the increased number of the classifiers. The inference of the classifiers based on the knowledge graph is valuable and credible. To avoid randomness, we perform three different domain adaptation tasks. We also test the influence of the number of the unknown classes, which is also shown in the above table. From the result showing above, we can come to the conclusion that our model shows a good performance on open set domain adaptation with zero-shot learning.

## 5 CONCLUSION

In this paper, we propose a model on open set domain adaptation with zero-shot learning. Our model not only makes good performance on the alignment of the domain gap but also can give detailed classification on the unknown classes. The ability of the further classification on the unknown classes improves the visual cognitive development ability of the robot, which is important for the robot working in a realistic environment. The experiments show that our model has a good performance on domain adaptation with zero-shot learning. In future work, we will test our model on the real scenario dataset, which is more suitable to the task of the visual cognitive development robot.

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
