# OpenReview forum: "Open Set Domain Adaptation with Zero-shot Learning on Graph"
_ICLR.cc/2022/Conference — ICLR 2022 Submitted_

### Official Review · Reviewer_fmyC · 2021-10-31

**Correctness:** 3
**Technical Novelty And Significance:** 3
**Empirical Novelty And Significance:** 2
**Recommendation:** 3
**Confidence:** 4

**Main Review:**


### Strengths

s1. Knowledge graph for learning classifiers: Unlike prior works, the paper aims at learning the classes present even in unknown bucket by using a zero-shot learning

s2. The paper tackles the open-set domain adaptation by generating classifiers that work well for unknown classes by utilizing Graph learning methods.

-------

### Weaknesses

w1. Motivation for a zero-shot learning and a clarification: Typically, Open-Set Domain-Adaptation focussed on "rejecting" unknown classes rather than modeling for the same explicitly [ref 1]. Therefore, these methods are useful in rejecting images even it they are not present in the pre-determined group of unknown classes. There are definitely some interesting use-cases such as incremental learning where the model adapts continually and learns new classes. So, it would be interesting to include some additional motivation for the zero-shot learning (on unknown classes) in real world with domain gap.

w2. Limited novelty of the proposed approach: The domain adaptation module and adversarial training procedure is not novel. This idea has been utilized in several papers. [ref 2, 4 etc.]. For ex, [ref 4] utilized $t$-parameter to separate the unknown classes from source class boundaries. So, the novelty of the paper is limited to utilization of GCN for unknown classes.

w3. Evaluation of the method and comparison against recent methods: The experiments are limited to  evaluation of the method on digits datasets such as MNIST. Although the results are encouraging, more experimental validation would be required across datasets that have richer attribute information (for Knowledge Graph). Further, the authors need to include additional comparisons from recent literature such as STA [ref 1], Inheritable models [ref 5] and closely related works such as [ref 3] (one-shot learning).

w4. Experimental protocol: It would be more interesting to consider an evaluation protocol that shows accuracy of unknown classes both as a "binary-classification" problem and "multi-class classification" problem.

-----

### Other aspects

a1. In my opinion, the paper considers an interesting problem but the idea of using exactly $n_t-n_s$ classes is unreasonable for practical reasons. First, there is already a domain shift among source and target distributions. Second, the target dataset is unlabeled. Therefore, it is hard to anchor out a strong distribution in this scenario. This problem has been addressed in a related work by utilizing prototypical learning [ref 3].

a2. As suggested in the above sections, the evaluation protocol should reflect the efficacy of method and its novel elements. This often requires providing additional metrics that have typically not been considered in literature.

a3. Including t-SNE plots would help the reader understand the qualitative performance of the proposed approach for the unknown classes.

-----

### Questions

q1. (Approach - Training strategy) Is there a prescribed way to alternate between the two stages for training the model to gain higher performance? Or is this a heuristic that yields higher performance.

q2. (Experiments - Settings section) How long does the model take to converge? The paper mentions that the model has been trained for 1200 epochs for digits dataset. I wonder if this is practical and if it would scale for larger datasets.

q3. (Experiments - Comparison section) It is not clear what the domain invariance means in the context of feature alignment for unknown classes.

-----

** Minor Typos **

m1. Closed-set domain adaptation and not "close"-set.

m2. There are few grammatical errors make the paper little hard  to follow.

----

**References**

[ref 1] Separate to Adapt: Open Set Domain Adaptation via Progressive Separation, CVPR'19

[ref 2] Adversarial Discriminative Domain Adaptation, CVPR'17

[ref 3] Class-Incremental Domain Adaptation, ECCV'20

[ref 4] Open Set Domain Adaptation by Backpropagation, ECCV'18

[ref 5] Towards inheritable models for open-set domain adaptation, CVPR'20

**Summary Of The Paper:**

The paper considers the problem of open-set domain adaptation where the target domain has additional group of unknown classes and domain-shift with source domain. One interesting aspect of the paper is that the method utilizes a Knowledge Graph for zero-shot learning on the unknown classes. Further, the method utilizes an adversarial learning approach to align the source and target domain.

**Summary Of The Review:**

I have provided the review for the paper in the previous sections. Although, the paper considers an interesting problem setting, more experimental validation would be required to judge its claims. Further, the novelty is limited to application of an already existing ideas. I'm happy to take the discussion with authors in the post-review phase to hear their thoughts. At the moment, the paper is simply not in acceptable state.

---

### Official Review · Reviewer_ymSC · 2021-11-02

**Correctness:** 2
**Technical Novelty And Significance:** 2
**Empirical Novelty And Significance:** Not applicable
**Recommendation:** 3
**Confidence:** 5

**Main Review:**

*Strong Points*
1. Their proposed setting is interesting and realistic.
2. The idea of using graph structure in the target domain sounds reasonable.

*Weak Points*
1. Though the high-level idea of leveraging a graph in the target domain sounds reasonable, their description of the method is not clear and not convincing enough. In Eq. 3, it is not clear where $w^{inf}$ comes from. Also, what are $M$ in Eq. 3 and Eq. 1? In addition, in Eq. 3, the objective is just to predict known classifier weights. But, why it is enough to get discriminative features for unknown samples?

2. Where is the attribute $v$ come from? This is missing in both method and experiments section.

3. Their evaluation is not enough to support the validity of their method. First, they perform experiments only on digits dataset, but this is clearly not enough. Second, they are lacking several baselines in open-set domain adaptation utilizing clustering methods such as "Do We Really Need to Access the Source Data" and "Universal Domain Adaptation through Self Supervision".

**Summary Of The Paper:**

They propose a new setting in open-set domain adaptation, where the goal is to classify known classes into their classes as well as cluster unknown classes well.  A difference from an existing open-set domain adaptation is that it does only require separating unknown instances from known ones whereas this paper aims to cluster unknown instances. For this goal, they propose a model for open set domain adaptation with zero-shot learning on the unknown classes. They combine adversarial learning to align the two domains, and the knowledge graph is introduced to generate the classifiers for the unknown classes with the employment of the graph convolution network (GCN).
They provide experiments on digits dataset and show the gain over baselines.

**Summary Of The Review:**

Considering the weak points, I recommend rejecting this paper. There are many missing parts in methods and they need more experiments to justify their ideas.

---

### Official Review · Reviewer_4Ng6 · 2021-11-03

**Correctness:** 3
**Technical Novelty And Significance:** 2
**Empirical Novelty And Significance:** 4
**Recommendation:** 5
**Confidence:** 4

**Main Review:**

This paper formulates a novel problem by stepping one step further from the open-set domain adaptation to classifying the unknown classes. Overall, the problem is interesting and can be practically useful. However, the work does not seem to be solid enough for a publication in ICLR.

Some of my concerns are as follows:

In the last sentence of 3.2, the authors claim "...but also provides mor general classifiers of the known ones.", is there any empirical evidence for this statement?

In eq (4) and (5), it is confusing to use t for both the "boundary" and the subsript for the target domain. And I don't understand the meaning of eq (4) and (5)

I don't see how the domain adaptation module can align two domains from the objective in eq (10) and (11). L_s make sure the classification accuracy on source-domain known classes, L_adv enables the separation of known and unknown classes in the target domain, and L_GCN learns a mapping from the knowledge graph node v to the classifier weight w. How is the domain adaptation done?

The knowledge graph should also be mentioned in the problem definition in section 3.1.

I believe the knowledge graph is important for good zero-shot learning, however, this information is not mentioned in the experiments.

Experiments on more datasets would be helpful to validate the benefit of the proposed method.

There are some language issues to fix, e.g., "To align the domain gap" should be "to bridge the domain gap" or something else;

**Summary Of The Paper:**

The paper formulates a novel problem in open-set domain adaptation and aims to classify the unknown classes in the target domain. This is different from the traditional open-set domain adaptation problem setting. To do this, additional knowledge of inter-class relations are employed and embedded so that knowledge learned from the shared classes can be transferred to the unknown classes.

**Summary Of The Review:**

In summary, the paper provides a solution to a practically useful problem but the novelty is limited and the presentation is not clear enough to understand.

---

### Decision · Program_Chairs · 2022-01-20

**Decision:**

Reject

**Comment:**

The paper addresses open-set DA, where samples from novel classes in the target domain get clustered
into new (unlabeled) classes. A key novelty in the learning setup is that it is assumed that one
has access to a knowledge graph over classes (both source and target). That KG is used for grouping
target samples into novel classes.

Reviewers were concerned that the method is not explained with sufficient
details and the experiments lacked comparisons with openset DA baselines.
No rebuttal was submitted.

The paper cannot be accepted to ICLR.